# Selected Aspects of Iodate and Iodosalicylate Metabolism in Lettuce Including the Activity of Vanadium Dependent Haloperoxidases as Affected by Exogenous Vanadium

**Sylwester Smoleń** [1,*], **Iwona Kowalska** [1], **Mariya Halka** [1], **Iwona Ledwożyw-Smoleń** [1], **Marlena Grzanka** [1], **Łukasz Skoczylas** [2], **Małgorzata Czernicka** [1] **and Joanna Pitala** [3]

[1] Department of Plant Biology and Biotechnology, Faculty of Biotechnology and Horticulture, University of Agriculture in Krakow, Al. 29 Listopada 54, 31-425 Krakow, Poland; iwona.kowalska@urk.edu.pl (I.K.); maryskagalka@gmail.com (M.H.); iwona.ledwozyw-smolen@urk.edu.pl (I.L.-S.); marlena.grzanka14@gmail.com (M.G.); malgorzata.czernicka@urk.edu.pl (M.C.)
[2] Department of Fruit, Vegetable and Mushroom Processing, Faculty of Food Technology, University of Agriculture in Krakow, Balicka 122, 30-149 Krakow, Poland; lukasz.skoczylas@urk.edu.pl
[3] Laboratory of Mass Spectrometry, Faculty of Biotechnology and Horticulture, University of Agriculture in Krakow, Al. 29 Listopada 54, 31-425 Krakow, Poland; joanna.pitala@urk.edu.pl
* Correspondence: sylwester.smolen@urk.edu.pl; Tel.: +48-12-662-52-39

**Abstract:** In marine algae, vanadium (V) regulates the cellular uptake of iodine (I) and its volatilization as $I_2$, the processes catalyzed by vanadium-dependent haloperoxidases (vHPO). Relationships between I and vanadium V in higher plants, including crop plants, have not yet been described. Little is known about the possibility of the synthesis of plant-derived thyroid hormone analogs (PDTHA) in crop plants. The activity of vHPO in crop plants as well as the uptake and metabolism of iodosalicylates in lettuce have not yet been studied. This study aimed to determine the effect of V on the uptake and accumulation of various forms of I, the metabolism of iodosalicylates and iodobenzoates and, finally, on the accumulation of T3 (triiodothyronine—as example of PDTHA) in plants. Lettuce (*Lactuca sativa* L. var. *capitata* 'Melodion' cv.) cultivation in a hydroponic Nutrient Film Technique (NFT) system was conducted with the introduction of 0 (control), 0.05, 0.1, 0.2, and 0.4 μM V doses of ammonium metavanadate ($NH_4VO_3$) in four independent experiments. No iodine treatment was applied in Experiment No. 1, while iodine compounds were applied at a dose of 10 μM (based on our own previous research) as $KIO_3$, 5-iodosalicylic acid (5-ISA) and 3,5-diiodosalicylic acid (3,5-diISA) in Experiment Nos. 2, 3 and 4, respectively. When lettuce was grown at trace amount of I in the nutrient solution, increasing doses of V contributed to the increase of (a) I content in roots, (b) I uptake by whole lettuce plants (leaves + roots), and (c) vHPO activity in leaves (for doses 0.05–0.20 μM V). Vanadium was mainly found in roots where the content of this element increased proportionally to its dose. The content of V in leaves was not modified by V introduced into the nutrient solution. We found that 5-ISA, 3,5-diISA and T3 were naturally synthesized in lettuce and its content increased when 5-ISA, 3,5-diISA were applied. Quantitative changes in the accumulation of organic metabolites (iodosalicylates and iodobenzoates) accumulation were observed, along with increased T3 synthesis, with its content in leaves exceeding the level of individual iodosalicylates and iodobenzoates. The content of T3 was not affected by V fertilization. It was concluded that iodosalicylates may participate in the biosynthesis pathway of T3—and probably of other PDTHA compounds.

**Keywords:** beneficial elements; iodobenzoates; iodosalicylates; plant-derived thyroid hormone analogs; T3; thyroid hormone; triiodothyronine; vanadium-dependent haloperoxidases

## 1. Introduction

Vanadium [1–3] and iodine [4,5] are included into the group of trace elements that are beneficial for human and animal organisms. In the human body, vanadium regulates the functioning of the following enzymes: Na/K ATPase, phosphotransferases, adenylate cyclase, and protein kinases [6,7]. Vanado-dependent enzymes exhibiting insulin-mimetic properties have also been described. Vanadium compounds may participate in the regulation of blood pressure, lipid profiles, and, along with iodine, the proper functioning of the thyroid gland [3,8].

Excessive intake of vanadium is toxic for humans [9] and farm animals [10] with the effect dependent on the dose, duration of the exposure, and chemical form of V. Vanadium pentoxide ($V_2O_5$) is the most toxic for human organisms, while the safest forms are vanadyl sulfate ($VOSO_4$) and metavanadate salts [11]. To date, the recommended daily allowance (RDA) for vanadium has not been determined [12]. The report of WHO only suggests that the tolerable upper intake level for adults is approximately 26 µg vanadium/kg body weight/day [6,7]. The lowest doses reported to cause adverse effects in humans were about 200 µg vanadium/kg/day, which is up to 1000 times higher than the approximate daily intake of that element, i.e., 0.2–0.3 µg/kg/day [13].

The role and importance of iodine as a trace element for humans and animal organisms is better recognized. Depending on the age, sex, and physiological condition, the iodine RDA for humans is between 90 and 200 µg I·day$^{-1}$[14]. Iodine is crucial for the synthesis of thyroid hormones: thyroxine (T4) and triiodothyronine (T3). Its deficiency causes a broad spectrum of disorders and diseases, from endemic goiter even to mental disability. Prenatal iodine deficiency leads to neurological fetal defects, including severe mental disability (cretinism) [15,16].

Vanadium and iodine are included into the group of beneficial elements for higher plants, i.e., that in low concentrations, they may stimulate plant growth and development. The effect of exogenous vanadium on plants strongly depends on its dose, chemical form (mainly oxidation state) and growth conditions, particularly with respect to the conditions of root system development [4,5,17]. Low concentrations of V (below 0.04 mg V·dm$^{-3}$) were revealed to improve chlorophyll biosynthesis and the uptake of P, K, Ca, and Mg by plants [18]. An increase of K and Mn accumulation after V application was noted in soybean leaves [19]. In the studies conducted by Sentíes-Herrera et al. [20], foliar application of 10 and 20 µM V (as $NH_4VO_3$) contributed to increased growth of sugar cane plants that were characterized by greater stem diameter and improved anatomic structure related to sugar accumulation. A positive effect of vanadium was also reflected by higher chlorophyll content in leaves as well as increased plant height, weight, number of leaves, and flowers of tomato plants [21].

The effect of iodine on plants is also strongly dependent on its dose, oxidation state, and cultivation conditions. In general, iodides (I$^-$) are more easily taken up than iodates (IO$_3^-$) but exhibit higher toxicity to plants [4]. Blasco et al. [22] observed that, in soilless cultivation of lettuce, application of I$^-$ in doses exceeding 80 µM I reduced plant biomass, while, for iodates, such an effect was not noted even at the dose of 240 µM I.

Plant preference towards the source of iodine varies depending on plant species. The order of iodine uptake by barley and pea is as follows—I$^-$ > CH$_2$ICOOH$^-$ > IO$_3^-$ > IO$_4^-$ [23], while for water spinach—CH$_2$ICOOH$^-$ > I$^-$ > IO$_3^-$ [24]. There is scarce information on the possibility of taking up organoiodine compounds with iodine bound to aromatic ring as in the case of iodosalicylates. Our previous research [25] revealed that 5-iodosalicylic acid can be absorbed by lettuce, effectively increasing iodine content in plant tissues. To date, no studies that have documented the effect of other iodosalicylates, such as 3,5-diiodosalicylic acid on lettuce plants, have been presented. The influence of that compound on young tomato plants was, however, tested [26].

Functioning and relations between iodine and vanadium in higher plants have not yet been recognized. Medrano-Macias et al. [4] informed about a possible interaction between these elements in terrestrial plants. However, plant response to inorganic or organic iodine as related to vanadium bioaccessibility is yet to be described. Adversely, in marine algae, that issue is widely studied.

In marine algae, enzymes belonging to the family of vanadium-dependent haloperoxidases (vHPO; vanadium-dependent bromo-, chloro- and mainly iodoperoxidases) participate in the process of iodine uptake from seawater into plant tissues [27]. Vanadium-dependent iodoperoxidase (vHIPO), present in the cell wall of *Laminaria digitata*, catalyzes the process of $I^-$ oxidation into hypoiodous (I) acid (HIO), which is further transformed into molecular $I_2$. These compounds (HIO and $I_2$) are more lipophillic than $I^-$, which facilitates its penetration through the cell membrane into the cytosol. The course of the processes of further reduction of HIO or $I_2$into $I^-$ in the apoplast is not known. vHIPO enzymes also participate in the process of volatilization of $I_2$ from cells to seawater. Hypoiodous acid (HIO) is also synthesized in the cell wall throughout the process, which is triggered in marine algae as a response to oxidative stress [27]. The activity of vHIPO enzymes was detected in various algae species, such as *Pelvetia canaliculata* [28], *Gracilaria fisheri* [29], and *Laminaria digitata* [30]. Nevertheless, the functioning of this enzyme in crop plants has not yet been sufficiently studied.

To date, plant fertilization with vanadium in hydroponic cultivation has not yet been commonly applied. That element is only slightly mentioned in the respective handbooks [31]. Our studies allow for filling the information gap on vanadium influence on crop plants, mainly with respect to iodine biofortification.

The aim of this study was to compare the effect of different doses of vanadium on iodine uptake (effectiveness of biofortification) by lettuce plants grown in the presence of $KIO_3$ and iodosalicylates: 5-iodosalicylic acid (5-ISA) or 3,5-diiodosalicylic acid (3,5-diISA) as source iodine compounds. The study was also directed at evaluating the activity of vHPO enzymes in lettuce leaves and roots under the influence of vanadium applied solely or together with iodine into the nutrient solution. It was also aimed at analyzing the chemical composition of plants affected by the application of vanadium as well as inorganic and organic iodine compounds.

## 2. Materials and Methods

### 2.1. Plant Material and Treatments

The hydroponic cultivation of lettuce *Lactuca sativa* L. var. *capitata* 'Melodion' cv. was conducted in an NFT (nutrient film technique) system located in a greenhouse of the Faculty of Biotechnology and Horticulture, University of Agriculture in Kraków (50°05′04.1″ N, 19°57′02.1″ E).

Four independent experiments (Experiment no. 1, 2, 3 and 4) were carried out according to the same procedure of lettuce cultivation (Table 1). Seeds were sown into 112-cell propagation trays $(330 \times 520 \times 40)$ mm with $(32 \times 32 \times 40)$ mm sized one cell filled with peat substrate mixed with sand (1:1 *v/v*). Seedlings of 4–5 true leaves were transplanted into the NFT system. Substrate was thoroughly rinsed from seedling root system with the use of tap water. Seedlings were placed into holes (spaced 25 cm apart) of styrofoam slabs filling NFT beds—a "dry hydroponic" method of cultivation without substrate. After transplanting, plants were watered for one minute every 5 min during the day between 5:00 a.m. and 7:00 p.m. and during the night between 1:00 a.m. and 2.00 a.m. The nutrient solution used for the cultivation contained the following amounts of macro- and microelements (mg·dm$^{-3}$): N 150, P 50, K 200, Mg 40, Ca 120, Fe 2, Mn 0.55, Zn 0.33, B 0.33, Cu 0.15, and Mo 0.05. At the beginning of lettuce cultivation in the NFT system, the EC (electrical conductivity) of nutrient solution of all treatments in all the experiments was 1.75 mS·cm$^{-1}$ and the pH of all nutrient solutions was adjusted to 5.70 with the use of 38% nitric acid in accordance with the recommendations of the Research Institute of Horticulture in Poland [32]. Measurements of pH and EC were carried out with the use of a pH-meter CP-505 and conducto-meter CC-505 (both by Elmetron Sp.j., Zabrze, Poland), respectively. The results of pH measurement were used to regulate the pH nutrient solution with the use of 38% nitric acid while readings of EC were stable during the whole cultivation period.

**Table 1.** Design, schedule and method of conducting experiments with lettuce cultivation in the NutrientFilm Technique (NFT) hydroponic system.

| Experimental Factor | Experiment No. 1 | Experiment No. 2 | Experiment No. 3 | Experiment No. 4 |
|---|---|---|---|---|
| Dose of vanadium as ammonium metavanadate ($\mu$M V nutrient solution). | 0 (control *) 0.05 0.1 0.2 0.4 | 0 (control *) 0.05 0.1 0.2 0.4 | 0 (control *) 0.05 0.1 0.2 0.4 | 0 (control *) 0.05 0.1 0.2 0.4 |
| Concentration and chemical form of iodine in nutrient solution (the same for each treatments in separateexperiment) ** | Control * (0.0204 $\mu$M I) | $KIO_3$ (potassium iodate) 10 $\mu$M (10 $\mu$M I) | 5-ISA (5-iodosalicylic acid) 10 $\mu$M (10 $\mu$M I) | 3,5-diISA (3,5-diiodo-salicylic acid) 10 $\mu$M (20 $\mu$M I) |
| Growing season | Autumn | Autumn | Autumn-winter | Autumn-winter |
| Seed sowing | 28 August 2018 | 28 August 2018 | 22 October 2018 | 22 October 2018 |
| Planting seedlings to NFT gutter | 20 September 2018 | 20 September 2018 | 16 November 2018 | 16 November 2018 |
| Iodine and vanadium application in rosette stage of plants. | 05 October 2018 | 05 October 2018 | 07 December 2018 | 07 December 2018 |
| Harvest of the lettuce heads (number of days after sowing/after planting seedlings to NFT gutter) | 05 November 2018 (69/46 days) | 05 November 2018 (69/46 days) | 08 January 2019 (78/53 days) | 08 January 2019 (78/53 days) |
| Parameters | | | | |
| Temperature for heating strategy (day/night) | 16 °C/10 °C | 16 °C/10 °C | 16 °C/10 °C | 16 °C/10 °C |
| Temperature for ventilation strategy (day/night) | 22 °C/15 °C | 22 °C/15 °C | 22 °C/15 °C | 22 °C/15 °C |
| Hours of natural light supplementation with 600-W high-pressure sodium lamps | 5.00–8.00 and 16.00–19.00 | 5.00–8.00 and 16.00–19.00 | 5.00–8.00 and 16.00–19.00 | 5.00–8.00 and 16.00–19.00 |

* control—trace concentration of iodine and vanadium in nutrient solution: 0.0204 $\mu$M I and 0.009 $\mu$M V of nutrient solution (I and V from tap water and fertilizers). ** Dose of iodine compounds for each vanadium treatment in Experiments Nos. 2–4.

Mineral nutrients were introduced into the solution with the use of the following fertilizers: calcium nitrate, monopotassium phosphate, potassium nitrate, potassium sulphate, magnesium nitrate (all produced/ distributed by Yara, Poland), and potassium chloride (ICL Speciality Fertilizer, City, Poland). Micronutrients were introduced in the form of multi-element fertilizer 'Mikro plus' (Intermag, Olkusz, Poland). The distinguishing factor of Experiment Nos. 1, 2, 3, and 4 was the chemical form of iodine applied with increasing doses of vanadium (Table 1). In each experiment, five doses of vanadium applied in the form of ammonium metavanadate ($NH_4VO_3$) were introduced into the nutrient solution: 0 (control) 0.05, 0.1, 0.2, and 0.4 $\mu$M V. For increasing V doses, the same concentration of iodine was applied. In Experiment No. 1, control iodine treatment was used, i.e., a trace amount of iodine in the nutrient solution. In subsequent experiments, the following iodine compounds were used: $KIO_3$ in Experiment No. 2; 5-iodosalicylic acid (5-ISA) in Experiment No. 3; and 3,5-diiodosalicylic acid (3,5-diISA) in Experiment No. 4.

Inorganic ($KIO_3$) and organic forms of iodine, i.e., 5-ISA and 3,5-diISA were applied in a dose of 10 $\mu$M calculated per molar mass of a whole compound. The dose was chosen based on own previously published [25,26] and unpublished results, including patent application P.410806; see Acknowledgments). According to that, the iodine dose was 10 $\mu$M I for $KIO_3$ and 5-ISA as well as 20 $\mu$M I for 3,5-diISA (Table 1). Iodine content in the control nutrient solution (**Experiment No. 1**) assayed by an ICP-MS QQQ (TQ ICP-MS ThermoFisher Scientific, Bremen, Germany) technique was 2.6 $\mu$g I·dm$^{-3}$ (0.0204 $\mu$M I)—trace amount delivered with tap water and mineral fertilizers. In each experiment, iodine and vanadium were first introduced at the stage of rosette with 4–5 true leaves. Prior to the introduction, respective amounts of 5-ISA and 3,5-diISA were dissolved in a small amount of water containing a few drops of 1M NaOH in order to increase its solubility.

The compartment in a greenhouse was equipped with 10 individual NFT sets with 650-dm³ nutrient solution containers, facilitating lettuce cultivation in recirculating hydroponics. Each NFT set consisted of three 5.5 m long beds. Each experiment was conducted separately in randomized block design with four repetitions within one NFT set. Plants were cultivated in four replications of 15 plants (60 plants per combination). In a single cultivation period, two experiments were performed, each with five levels of vanadium application into the nutrient solution (ten combinations in total, each combination in an individual NFT set). Experiment Nos. 1 and 2 were carried out in the autumn season and Experiment Nos. 3 and 4 in the autumn–winter season of 2018. Cultivation in both terms (autumn and autumn–winter season) was done with the same microclimate conditions regulated with the use of a greenhouse climate computer control system (Table 1). Natural light was supplemented between 5:00 a.m. and 8:00 a.m. as well as 4:00 p.m. and 7:00 p.m. with the use of 600-W high-pressure sodium lamps, which allowed for obtaining a 14h day/10h night photoperiod. During the harvest, the biomass of lettuce roots and heads were evaluated. Chemical analyses were performed using the collected plant material i.e., five heads and all the roots from plants from each replication.

### 2.2. Activity of Vanadium-Dependent Haloperoxidases (vHPO)

An analysis of a total activity of vHPO enzymes in the fresh samples of lettuce roots and leaves was conducted. No information has been found in the literature on measuring the activity of vHPO enzymes in higher plants. Therefore, the analysis was performed based on the methods used for marine algae [28,29,33,34]. The adapted analytical method allowed us to measure the total activity of vanadium-dependent haloperoxidase enzymes (vHPO) in root and leaf tissues of lettuce.

Roots and leaves of lettuce were washed in tap and distilled water, dried with the use of laboratory paper and cut into small fragments. The samples of 2.5 g were weighted into 30 mL Falcon tubes and homogenized with 5 mL ice-cold 20 mM Tris HCl buffer (pH 8.5) containing 1% PVP-40 [29]. Homogenized plant material was transferred into centrifuge tubes and centrifuged for 15 min at 4500 rpm, 2 °C. The supernatant was collected, transferred to fresh centrifuge tubes, and further centrifuged for 10 min at 12,000 rpm, 2 °C. An amount of 2 mL of supernatant was collected from the Eppendorf tubes for the enzymatic activity analysis.

The enzymatic activity of vHPO was assayed spectrophotometrically with the use of a Hitachi U2900 Spectrophotometer (Hitachi High Technologies Corporation, Tokyo, Japan). The total volume of reaction mixture was 3 mL and contained (according to the order of mixing): (a) 1.3 mL of Tris-HCl pH 8.5; (b) 0.1 mL of 1 mM KI; (c) 0.25 mL of 1 mM $NH_4VO_3$; (d) 0.5 mL of extract (or double distilled $H_2O$ for blank); and (e) 0.1 mL of 100 μM bromothymol blue (TB). The mixture was vortexed for a few seconds, transferred into the cuvette and then 0.75 mL of 10 mM $H_2O_2$ was added to initiate the reaction. All reagents, i.e., KI, $NH_4VO_3$, TB, and $H_2O_2$ were dissolved in Tris-HCl pH 8.5. The absorbance was read at 0 and 20 min time at 620 nm [33]. The activity of vanadium-dependent haloperoxidase was calculated based on the increase of absorbance after 20 min and expressed as $U \cdot mg^{-1} \cdot min^{-1}$ protein. The protein content was assayed according to the Lowry method with bovine serum albumin as a standard [35].

### 2.3. Analysis of Dry Samples of Roots and Leaves

Each lettuce head was cut in half (the leaves from each growth stage was collected), mixed within the replication, frozen at −20 °C and lyophilized with the use of a Christ Alpha 1-4 lyophilizer (Martin Christ Gefriertrocknungsanlagen GmbH, Germany). Lettuce roots were prepared accordingly. Samples of lyophilized leaves and roots were ground in a laboratory grinder FRITSCH Pulverisette 14 (FRITSCH GmbH, Weimar, Germany) and stored in tightly closed polyethylene bags (at room temperature) until the analysis.

***Iodine determination.*** In order to evaluate the content of iodine in root and leaf samples, the alkaline extraction of samples with tetramethylammonium hydroxide (TMAH) was conducted according to the procedure described in previous work [36]: 0.5 g air-dried leaf or root samples, 10 mL double-distilled

water and 1 mL of 25% TMAH (Sigma-Aldrich, St. Louis, MO, USA) were put into 30mLFalcon tubes. After mixing, samples were incubated for 3 h at 90 °C. After incubation, samples were cooled to a temperature of approximately 20 °C and filled to 30 mL with double-distilled water. After mixing, samples were centrifuged for 15 min at 4500 rpm. The measurements of iodine content using an ICP-MS triple quadruple spectrometer (TQ ICP-MS Thermo Fisher Scientific) were conducted in the supernatant without decanting [37].

*Vanadium determination.* An analysis of vanadium content was conducted with the use of ICP-OES technique (using an ICP-OES Prodigy Spectrometer, Leeman Labs, New Hampshire, MA, USA) after microwave digestion in 65% super pure $HNO_3$[38]. The 0.5 g plant samples were placed in 55 mL TFM (TFM—Modified Poly-Tetra-Fluoro-Ethylene (PTFE)) vessels and were digested in a 10 mL 65% super pure $HNO_3$ (Merck Whitehouse, Station, NJ, USA) in a CEM MARS-5 Xpress (CEM World Headquarters, Matthews, NC, USA) microwave digestion system. The following procedure was applied: 15 min time needed to achieve a temperature of 200 °C and 20 min maintaining this temperature. After cooling, the samples were quantitatively transferred to 25 mL graduated flasks with redistilled water.

*Determination of salicylic acid (SA), benzoic acid (BeA), iodosalicylates, iodobenzoates and triiodothyronine (T3).* The content of SA, 5-ISA, 3,5-diISA, 2-iodobenzoic acid (2-IBeA), 4-iodobenzoic acid (4-IBeA), 2,3,5-triiodobenzoic acid (2,3,5-triIBeA), and T3 was analyzed in the root and leaf samples. The following extraction procedure was applied: 50 mg of plant material was placed in 7 mL polypropylene tubes, 5 mL of 76% ethanol containing 50 ng·mL$^{-1}$ of deutered salicylic acid (SA-d4, Sigma-Aldrich). Samples were vortexed and subjected to 1-h ultrasound-assisted extraction at 50 °C. Samples were then centrifuged (5 min, 4500 RPM) and supernatants filtered with the use of nylon 0.22 μm syringe filters (FilterBio NY Syringe Filter, Phenomenex, Torrance, CA, USA). The measurements of SA, BeA, 5-ISA, 3,5-diISA, 2-IBeA, 4-IBeA, and 2,3,5-triIBeA were determined by an LC-MS/MS technique (Ultimate 3000, Thermo Scientific, QTrap 4500, Sciex) according to the previously described procedure [39]. Chromatographic separation was carried out on a Luna 3 μm Phenyl-Hexyl 100 Å (150 mm × 3 mm, i.d. 3 μm) column (Phenomenex, Phenomenex, Torrance, CA, USA) with the following mobile phases: A—water with formic acid 0.3% (at the beginning 60%); B—acetonitrile with formic acid 0.3% (40%). After 2 min, the proportions of the mobile phase were increased linearly up to obtain 98% phase B at 8 min and held for 4 min. The starting proportions were restored over a 3 min period after the 15-min analysis. The injection volume was 10 μL. The mobile phase was directed to MS ion source between 1 and 14 min of the separation. For detection, electrospray ionization (ESI) in negative ion mode was used. Tandem mass spectrometry MS/MS was used for quantitative studies. 136.8/93.1, 120.9/76.9, 262.9/126.7, 388.8/126.7, 246.9/126.6, 246.9/144.7, 498.7/454.4, 671.8/350.2 and 141/96.8 transitions were monitored for SA, BeA, 5-ISA, 3,5-diISA, 2-IBeA, 4-IBeA, 2,3,5-triIBeA, T3 sodium salt and for SA-d4, respectively. The LC-MS/MS system was controlled using Analyst 1.7 with HotFix 3 software, which was also used for data processing.

### 2.4. Statistical Analysis

All data were statistically verified using the ANOVA module of the Statistica 12.0 PL program (https://www.tibco.com/products/data-science, StatSoft Inc., Tulsa, OK 74104, USA) at significance level of $p < 0.05$. In the case of significant effects, homogenous groups were distinguished on the basis of a Tukey test. The obtained results were statistically verified separately for each of the four experiments.

## 3. Results

### 3.1. Plant Biomass

The weights of lettuce heads from Experiment Nos. 3 and 4 were approximately 7.5 times lower than those from Experiment Nos. 1 and 2 (Figure 1 and Table 2). This indicates that 5-ISA and 3,5-diISA

applied in a 10 μM dose similarly inhibited the growth and development of plants as compared to the same concentration of $KIO_3$.

Increasing doses of vanadium in the nutrient solution had no significant effect on head and root biomass in lettuce grown in the control as well $KIO_3$ or 5-ISA combinations (Experiment Nos. 1, 2, and 3, Table 2). Only the application of the highest doses of vanadium 0.40 μM V with 3,5-diISA caused a 27.1% reduction of lettuce head biomass, yet the effect was not statistically significant and also not observed in roots (Experiment 4, Table 2). Application into the nutrient solution of 0.10 and 0.20 μM V with 3,5-diISA significantly improved root biomass of lettuce, respectively, by 56.8% and 43.7%.

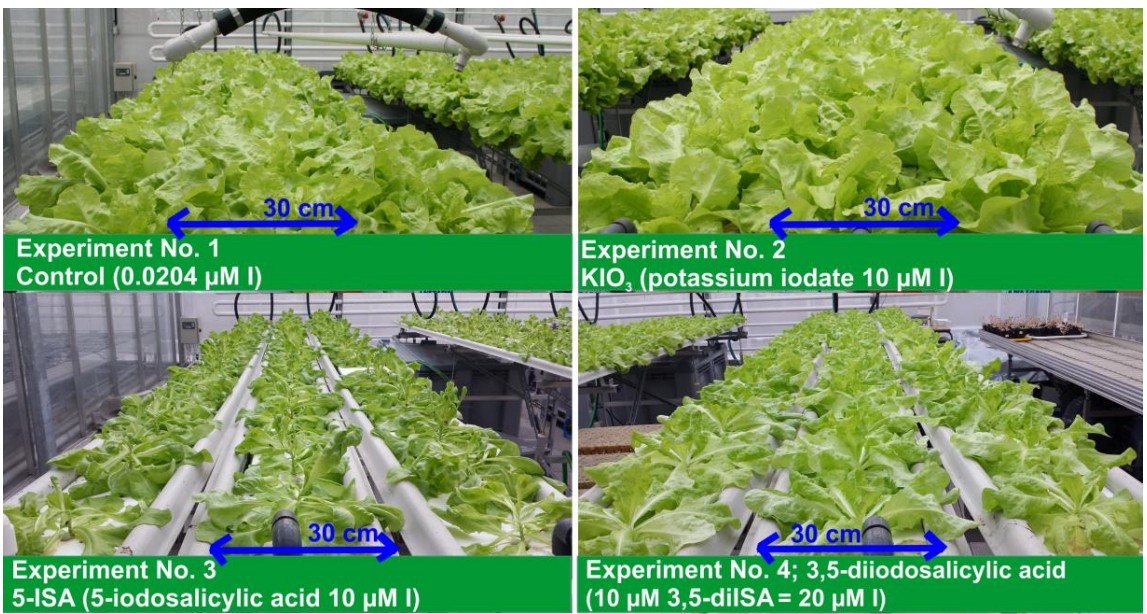

**Figure 1.** Lettuce plants prior harvest from Experiment Nos. 1, 2, 3 and 4.

**Table 2.** Effects of $KIO_3$, 5-iodosalicylic acid (5-ISA), 3,5-diiodosalicylic acid (3,5-diISA)andvanadium application on lettuce biomass.

| Exp. No. | Treatments | Leaf Biomass (Lettuce Head, g) | Root Biomass from One Plant (g) | Whole Plant Biomass (Roots + Leaves, g) |
|---|---|---|---|---|
| 1 | Control | 238.5 ± 6.06a | 22.6 ± 0.72a | 261.1 ± 6.60a |
| | 0.05 μM V | 253.1 ± 12.78a | 25.1 ± 1.60a | 278.3 ± 13.94a |
| | 0.10 μM V | 267.6 ± 5.21a | 27.1 ± 1.14a | 294.8 ± 4.62a |
| | 0.20 μM V | 269.7 ± 3.51a | 27.4 ± 1.05a | 297.0 ± 3.04a |
| | 0.40 μM V | 247.4 ± 18.76a | 24.4 ± 1.59a | 271.8 ± 19.68a |
| 2 | Cont. + $KIO_3$ | 254.4 ± 3.37a | 23.8 ± 0.63a | 278.1 ± 3.43a |
| | 0.05 μM V + $KIO_3$ | 233.8 ± 3.74a | 23.6 ± 0.86a | 257.3 ± 3.71a |
| | 0.10 μM V + $KIO_3$ | 246.7 ± 14.74a | 25.5 ± 1.26a | 272.2 ± 15.03a |
| | 0.20 μM V + $KIO_3$ | 252.5 ± 8.37a | 24.8 ± 1.58a | 277.3 ± 9.59a |
| | 0.40 μM V + $KIO_3$ | 245.6 ± 13.26a | 21.4 ± 1.79a | 267.0 ± 14.94a |
| 3 | Cont. + 5-ISA | 29.50 ± 1.14a | 4.50 ± 0.46a | 34.0 ± 1.58a |
| | 0.05 μM V + 5-ISA | 29.25 ± 1.90a | 4.88 ± 0.47a | 34.1 ± 2.37a |
| | 0.10 μM V + 5-ISA | 28.50 ± 1.85a | 4.38 ± 0.55a | 32.9 ± 2.11a |
| | 0.20 μM V + 5-ISA | 27.33 ± 1.85a | 4.25 ± 0.60a | 31.6 ± 2.16a |
| | 0.40 μM V + 5-ISA | 32.38 ± 0.80a | 4.13 ± 0.24a | 36.5 ± 0.79a |
| 4 | Cont. + 3,5-diISA | 40.94 ± 3.36ab | 3.59 ± 0.30ab | 44.5 ± 3.64ab |
| | 0.05 μM V + 3,5-diISA | 39.38 ± 1.14ab | 4.69 ± 0.18bc | 44.1 ± 1.16ab |
| | 0.10 μM V + 3,5-diISA | 43.13 ± 4.29ab | 5.63 ± 0.26c | 48.8 ± 4.51b |
| | 0.20 μM V + 3,5-diISA | 49.22 ± 2.98b | 5.16 ± 0.47c | 54.4 ± 3.20b |
| | 0.40 μM V + 3,5-diISA | 29.84 ± 2.82a | 3.13 ± 0.26a | 33.0 ± 3.05a |

Means in the column followed by different letters separately for each experiment differ significantly at $p < 0.05$ ($n = 4$).

### 3.2. Iodine Uptake and Accumulation by Lettuce Plants

A various effect of vanadium on iodine content in leaves and roots of lettuce was noted depending on the chemical form of iodine present in the nutrient solution (Table 3).

Each dose of vanadium significantly increased iodine content in roots of plants grown in the control and KIO$_3$ combinations (Experiment Nos. 1 and 2, Table 3); for the trace amount of iodine in the nutrient solution, iodine level in roots increased proportionally to the applied level of vanadium (Experiment No. 1). Iodine content in leaves of plants from Experiment No. 1 was higher when lower doses of V were applied i.e., 0.05 and 0.10 μM V.

In comparison to the control, application of all the doses of vanadium caused a significant decrease in iodine content in the leaves of plants fertilized with KIO$_3$ (Experiment No. 2, Table 3) as well as in the leaves and roots of lettuce from 5-ISA combinations (Experiment No. 3). For 3,5-diISA, only its highest dose, i.e., 0.40 μM V reduced iodine content in roots (Experiment No.4). At the same time, no clear effect of vanadium doses on iodine content was noted in leaves of lettuce from that experiment.

To sum up, only for trace amounts of iodine in the nutrient solution in Experiment No. 1 did additional application of increasing doses of vanadium increase iodine uptake for a single head(all leaves from plants), roots, and/or whole plants of lettuce (Table 4). When iodine was applied in the form of KIO$_3$, 5-ISA, and 3,5-diISA, the application of the highest dose of vanadium decreased iodine uptake by a whole lettuce plant. Additionally, the application of 0.10 and 0.20 μM V + 3,5-diISA increased iodine uptake by lettuce heads, roots, and whole plants (head+roots). In all experiments, the two lowest vanadium doses, i.e.,0.05 and 0.10 μM V, decreased the vanadium uptake by the roots as compared to respective control combinations.

**Table 3.** Effect of KIO$_3$, 5-iodosalicylic acid (5-ISA), 3,5-diiodosalicylic acid (3,5-diISA) and vanadium application on the content of iodine in leaves and roots of lettuce plants.

| Exp. No. | Treatments | Content of Iodine (mg I·kg$^{-1}$ D.W.) | |
| :---: | :---: | :---: | :---: |
| | | in Leaves | in Roots |
| 1 | Control | 0.21 ± 0.006a | 2.91 ± 0.440a |
| | 0.05 μM V | 0.51 ± 0.009d | 10.46 ± 0.149b |
| | 0.10 μM V | 0.55 ± 0.005e | 12.89 ± 0.252c |
| | 0.20 μM V | 0.29 ± 0.003b | 15.21 ± 0.156d |
| | 0.40 μM V | 0.32 ± 0.002c | 17.43 ± 0.345e |
| 2 | Cont. + KIO$_3$ | 10.56 ± 0.184d | 33.51 ± 0.565a |
| | 0.05 μM V + KIO$_3$ | 8.01 ± 0.044b | 45.77 ± 1.256c |
| | 0.10 μM V + KIO$_3$ | 9.14 ± 0.140c | 52.88 ± 1.117d |
| | 0.20 μM V + KIO$_3$ | 8.65 ± 0.285bc | 35.41 ± 0.163b |
| | 0.40 μM V + KIO$_3$ | 6.99 ± 0.146a | 46.80 ± 1.019c |
| 3 | Cont. + 5-ISA | 286.77 ± 3.462b | 1 111.30 ± 20.512a |
| | 0.05 μM V + 5-ISA | 243.50 ± 0.583a | 1 029.87 ± 4.949b |
| | 0.10 μM V + 5-ISA | 247.68 ± 7.197a | 941.42 ± 11.650a |
| | 0.20 μM V + 5-ISA | 233.31 ± 1.982a | 919.30 ± 18.200a |
| | 0.40 μM V + 5-ISA | 238.82 ± 1.480a | 943.14 ± 7.584a |
| 4 | Cont. + 3,5-diISA | 9.68 ± 0.084b | 646.79 ± 54.627bc |
| | 0.05 μM V + 3,5-diISA | 8.82 ± 0.036a | 678.97 ± 34.856bc |
| | 0.10 μM V + 3,5-diISA | 12.11 ± 0.127d | 764.66 ± 18.047c |
| | 0.20 μM V + 3,5-diISA | 9.50 ± 0.329ab | 603.05 ± 19.232ab |
| | 0.40 μM V + 3,5-diISA | 11.23 ± 0.135c | 466.95 ± 36.763a |

Means in the column followed by different letters separately for each experiment differ significantly at $p < 0.05$ ($n = 4$).

**Table 4.** Iodine and vanadium uptake by single lettuce head (all leaves from plants), roots as well as whole plant (head + roots).

| Exp. No. | Treatments | Iodine Uptake | | | Vanadium Uptake | | |
|---|---|---|---|---|---|---|---|
| | | by Single Head (µg I·head$^{-1}$) | by Roots (µg I roots·plant$^{-1}$) | by Whole Plant (Head + Roots) (µg I plant$^{-1}$) | by Single Head (µg V·head$^{-1}$) | by Roots (µg V roots·plant$^{-1}$) | by Whole Plant (Head + Roots) (µg V plant$^{-1}$) |
| 1 | Control | 1.85 ± 0.05a | 1.92 ± 0.1a | 3.76 ± 0.1a | 8.54 ± 0.2a | 1.95 ± 0.1a | 10.49 ± 0.2a |
| | 0.05 µM V | 4.99 ± 0.09c | 8.28 ± 0.1b | 13.27 ± 0.1b | 9.44 ± 0.1bc | 4.26 ± 0.1b | 13.70 ± 0.1b |
| | 0.10 µM V | 5.60 ± 0.04d | 12.61 ± 0.2c | 18.20 ± 0.2d | 10.09 ± 0.2c | 9.34 ± 0.1c | 19.43 ± 0.1c |
| | 0.20 µM V | 3.03 ± 0.03b | 13.95 ± 0.1d | 16.98 ± 0.1c | 9.93 ± 0.2c | 10.98 ± 0.7d | 20.91 ± 0.6c |
| | 0.40 µM V | 3.15 ± 0.02b | 14.37 ± 0.3d | 17.52 ± 0.2cd | 8.76 ± 0.2ab | 19.10 ± 0.1e | 27.86 ± 0.1d |
| 2 | Cont. + KIO$_3$ | 106.0 ± 1.8c | 24.5 ± 0.4a | 130.5 ± 2.1d | 9.51 ± 0.1bc | 1.19 ± 0. 1a | 10.7 ± 0.1a |
| | 0.05 µM V + KIO$_3$ | 75.4 ± 0.4a | 33.7 ± 0.9b | 109.2 ± 0.8ab | 8.43 ± 0.1a | 3.86 ± 0.1b | 12.3 ± 0.1a |
| | 0.10 µM V + KIO$_3$ | 84.5 ± 1.3b | 36.9 ± 0.8c | 121.4 ± 1.9c | 8.57 ± 0.2a | 6.71 ± 0.1c | 15.3 ± 0.3b |
| | 0.20 µM V + KIO$_3$ | 85.5 ± 2.8b | 31.3 ± 0.1b | 116.8 ± 2.7bc | 8.81 ± 0.1ab | 10.36 ± 0.1d | 19.2 ± 0.2c |
| | 0.40 µM V + KIO$_3$ | 70.0 ± 1.5a | 31.8 ± 0.7b | 101.7 ± 1.3a | 9.59 ± 0.1c | 16.14 ± 1.2e | 25.7 ± 1.2d |
| 3 | Cont. + 5-ISA | 502.8 ± 6.0b | 218.0 ± 4.0b | 720.9 ± 8.8ab | 2.73 ± 0.1a | 0.11 ± 0.1a | 2.83 ± 0.1a |
| | 0.05 µM V + 5-ISA | 469.4 ± 1.1a | 256.3 ± 1.2c | 725.7 ± 2.2b | 3.03 ± 0.1ab | 0.48 ± 0.1b | 3.51 ± 0.1a |
| | 0.10 µM V + 5-ISA | 511.6 ± 14.8b | 221.7 ± 2.7b | 733.4 ± 14.5b | 3.58 ± 0.1b | 1.20 ± 0.1c | 4.78 ± 0.1b |
| | 0.20 µM V + 5-ISA | 457.8 ± 3.8a | 229.0 ± 4.5b | 686.8 ± 2.5a | 3.44 ± 0.1b | 4.03 ± 0.1d | 7.47 ± 0.1c |
| | 0.40 µM V + 5-ISA | 531.8 ± 3.3b | 171.5 ± 1.5a | 703.3 ± 2.7a | 3.67 ± 0.1b | 6.19 ± 0.1e | 9.86 ± 0.3d |
| 4 | Cont. + 3,5-diISA | 28.1 ± 0.2b | 164.8 ± 13.9b | 192.8 ± 13.8b | 4.68 ± 0.1bc | 0.33 ± 0.1a | 5.01 ± 0.1a |
| | 0.05 µM V + 3,5-diISA | 23.9 ± 0.1a | 226.2 ± 11.6c | 250.1 ± 11.7c | 4.13 ± 0.2b | 1.63 ± 0.1b | 5.76 ± 0.2a |
| | 0.10 µM V + 3,5-diISA | 37.1 ± 0.4d | 322.9 ± 7.6d | 360.3 ± 8.0d | 4.83 ± 0.2c | 3.17 ± 0.1c | 8.00 ± 0.2b |
| | 0.20 µM V + 3,5-diISA | 31.7 ± 1.1c | 223.6 ± 7.1c | 256.8 ± 6.9c | 6.06 ± 0.1d | 6.76 ± 0.1d | 12.82 ± 0.1c |
| | 0.40 µM V + 3,5-diISA | 22.0 ± 0.2a | 107.4 ± 8.5a | 127.4 ± 8.3a | 3.40 ± 0.1a | 8.63 ± 0.1e | 12.03 ± 0.2c |

Means in the column followed by different letters separately for each experiment differ significantly at $p < 0.05$ ($n = 8$).

## 3.3. Vanadium Uptake and Accumulation by Lettuce.

In each experiment, the level of vanadium in lettuce leaves was not affected by exogenous application of that element into the nutrient solution (Table 5). However, a gradual and significant increase of vanadium content in roots caused by increasing vanadium level in the nutrient solution was noted in all four experiments. The average content of vanadium in leaves was lower than in roots—approximately by 91.0%, 91.4%, 86.4%, and 88.1%, respectively, for Experiment Nos. 1, 2, 3, and 4.

**Table 5.** Effect of KIO$_3$, 5-iodosalicylic acid (5-ISA), 3,5-diiodosalicylic acid (3,5-diISA) and vanadium application on the content of vanadium in leaves and roots of lettuce plants.

| Exp. No. | Treatments | Content of Vanadium (mg V·kg$^{-1}$ D.W.) | |
|---|---|---|---|
| | | in Leaves | in Roots |
| 1 | Control | 0.95 ± 0.024ab | 2.96 ± 0.027a |
| | 0.05 µM V | 0.96 ± 0.011ab | 5.38 ± 0.035b |
| | 0.10 µM V | 0.99 ± 0.014b | 9.55 ± 0.146c |
| | 0.20 µM V | 0.95 ± 0.018ab | 11.98 ± 0.802d |
| | 0.40 µM V | 0.90 ± 0.021a | 23.18 ± 0.111e |
| 2 | Cont. + KIO$_3$ | 0.95 ± 0.015a | 1.64 ± 0.014a |
| | 0.05 µM V + KIO$_3$ | 0.89 ± 0.019a | 5.24 ± 0.115b |
| | 0.10 µM V + KIO$_3$ | 0.93 ± 0.024a | 9.45 ± 0.169c |
| | 0.20 µM V + KIO$_3$ | 0.89 ± 0.018a | 13.51 ± 0.115d |
| | 0.40 µM V + KIO$_3$ | 0.96 ± 0.009a | 23.76 ± 1.846e |
| 3 | Cont. + 5-ISA | 1.55 ± 0.018a | 0.56 ± 0.009a |
| | 0.05 µM V + 5-ISA | 1.57 ± 0.077a | 1.94 ± 0.033b |
| | 0.10 µM V + 5-ISA | 1.73 ± 0.029a | 5.11 ± 0.072c |
| | 0.20 µM V + 5-ISA | 1.75 ± 0.039a | 16.19 ± 0.169d |
| | 0.40 µM V + 5-ISA | 1.25 ± 0.132a | 34.05 ± 0.581e |
| 4 | Cont. + 3,5-diISA | 1.62 ± 0.027 a | 1.29 ± 0.081a |
| | 0.05 µM V + 3,5-diISA | 1.53 ± 0.079 a | 4.90 ± 0.051b |
| | 0.10 µM V + 3,5-diISA | 1.58 ± 0.083 a | 7.50 ± 0.059c |
| | 0.20 µM V + 3,5-diISA | 1.82 ± 0.022 a | 18.23 ± 0.053d |
| | 0.40 µM V + 3,5-diISA | 1.73 ± 0.045 a | 37.50 ± 0.582e |

Means in the column followed by different letters separately for each experiment differ significantly at $p < 0.05$ ($n = 4$).

It can therefore be stated that increasing doses of ammonium metavanadate in the nutrient solution proportionally increased vanadium uptake by the roots from a single plant as well as by whole plants (Table 4). The level of vanadium translocation into leaves was not directly affected by vanadium dose.

### 3.4. Effect of Iodine and Vanadium on vHPO Activity in Lettuce Plants

Depending on the chemical form of applied iodine, various effects of vanadium application on vHPO activity in lettuce leaves and roots were noted (Table 6). A significant increase of vHPO activity in lettuce leaves caused by increasing doses of vanadium was noted only in plants grown in the presence of trace amounts of iodine in the nutrient solution (Experiment No.1). Vanadium doses of 0.05, 0.1, and 0.2 μM V applied to plants fertilized with $KIO_3$ contributed to an approximately four-time reduction of vHPO activity in leaves, while for the highest dose (0.4 μM V), vHPO activity was comparable to the control (Experiment No. 2). Only the highest dose of vanadium, i.e., 0.40 μM V,decreased vHPO activity in the leaves of lettuce grown with 3,5-diISA as an iodine source (Experiment no. 4). No changes in the activity of vHPO in roots were observed in all the conducted experiments (Table 6).

**Table 6.** Effect of $KIO_3$, 5-iodosalicylic acid (5-ISA), 3,5-diiodosalicylic acid (3,5-diISA) and vanadium application on the activity of vanadium-dependent haloperoxidase(vHPO).

| Exp. No. | Treatments | vHPO Activity (U·μg$^{-1}$ protein) | |
| --- | --- | --- | --- |
| | | in Leaves | in Roots |
| 1 | Control | 0.298 ± 0.021a | 3.110 ± 0.036a |
| | 0.05 μM V | 0.505 ± 0.040ab | 5.163 ± 0.012a |
| | 0.10 μM V | 0.728 ± 0.010bc | 5.413 ± 0.007a |
| | 0.20 μM V | 1.243 ± 0.011d | 3.690 ± 0.006a |
| | 0.40 μM V | 0.903 ± 0.040c | 3.548 ± 0.014a |
| 2 | Cont. + $KIO_3$ | 1.213 ± 0.030bc | 3.000 ± 0.043ab |
| | 0.05 μM V + $KIO_3$ | 0.310 ± 0.010a | 2.840 ± 0.027ab |
| | 0.10 μM V + $KIO_3$ | 0.465 ± 0.022ab | 1.645 ± 0.009a |
| | 0.20 μM V + $KIO_3$ | 0.238 ± 0.020a | 2.883 ± 0.030ab |
| | 0.40 μM V + $KIO_3$ | 1.908 ± 0.010c | 5.105 ± 0.010b |
| 3 | Cont. + 5-ISA | 1.585 ± 0.040b | 0.750 ± 0.041a |
| | 0.05 μM V + 5-ISA | 0.910 ± 0.020ab | 0.484 ± 0.078a |
| | 0.10 μM V + 5-ISA | 1.385 ± 0.052b | 0.593 ± 0.051a |
| | 0.20 μM V + 5-ISA | 1.088 ± 0.01b | 0.590 ± 0.015a |
| | 0.40 μM V + 5-ISA | 0.568 ± 0.020a | 0.530 ± 0.052a |
| 4 | Cont. + 3,5-diISA | 1.066 ± 0.027b | 0.397 ± 0.061a |
| | 0.05 μM V + 3,5-diISA | 0.901 ± 0.008b | 0.313 ± 0.060a |
| | 0.10 μM V + 3,5-diISA | 0.587 ± 0.005ab | 0.342 ± 0.063a |
| | 0.20 μM V + 3,5-diISA | 0.499 ± 0.014ab | 0.347 ± 0.059a |
| | 0.40 μM V + 3,5-diISA | 0.263 ± 0.005a | 0.421 ± 0.080a |

Means in the column followed by different letters separately for each experiment differ significantly at $p < 0.05$ ($n = 8$).

### 3.5. The Content of BeA, SA, Iodosalicylates, Iodobenzoates, and T3 in Lettuce Plants

Only in plants fertilized with 5-ISA did the tested vanadium doses increase the level of 5-ISA in lettuce rootssignificantly, at the same time decreasing its content in leaves as compared to the control, i.e., 5-ISA applied without vanadium (Experiment No. 3; Table 7.). Vanadium applied in doses of 0.1, 0.2 and 0.4 μM V decreased the level of SA and increased the content of 5-ISA in the leaves of plants grown in the presence of 3,5-diISA (Experiment No. 4).

Irrespective of vanadium dose, the content of 5-ISA, 3,5-diISA, 2-IBeA, 4-IBeA, and 2,3,5-triIBeA in the leaves of plants grown in the presence of trace amount of iodine (Experiment No. 1) was a few times higher than in plants treated with $KIO_3$ (Experiment No. 2). Moreover, the content of T3 in the roots of plants from the experiments with trace iodine amount and $KIO_3$ (No. 1 and 2) was approximately

20 times higher than in the leaves. The content of T3 in the leaves and roots from 5-ISA and 3,5-diISA combinations remained at a similar level. At the same time, the content of T3 in leaves from these plants was from 6 to 9 times higher than in lettuce grown in the presence of trace iodine amounts and $KIO_3$ in the nutrient solution (Experiment Nos. 1 and 2). The highest amount of SA, 2-IBeA, 4-IBeA, and 2,3,5-triIBeA in leaves was noted in plants non-fertilized with iodine (Experiment 1, Table 7).

The content of BeA in the leaves of lettuce not fertilized with iodine (Experiment No. 1) was similar to that of SA, while, in plants fertilized with $KIO_3$ (Experiment No.2), it was two times higher.

The application of 5-ISA and 3,5-diISA resulted in the significant increase in the content of 5-ISA and 3,5-diISA in lettuce leaves and roots (as compared to plants grown in the presence of trace amounts of iodine and $KIO_3$), which suggests a possible translocation of these compounds from roots to leaves. It was also found that the application of 5-ISA (Experiment No. 3) increased the content of SA and 3,5-diISA only in roots (Table 7).

**Table 7.** Benzoic acid (BeA), salicylic acid (SA), 5-iodosalicylic acid (5-ISA), 3,5-diiodosalicylic acid (3,5-diISA), 2-iodobenzoic acid (2-IBeA), 4-iodobenzoic acid (4-IBeA), 2,3,5-triiodobenzoic acid (2,3,5-triIBeA), and triiodothyronine (T3) in leaves and roots of lettuce plants.

| Exp. No. | Part of Plants | Treatments | (mg·kg⁻¹ D.W.) | | | | | | | |
|---|---|---|---|---|---|---|---|---|---|---|
| | | | BeA | SA | 5-ISA | 3,5-diISA | 2-IBeA | 4-IBeA | 2,3,5-TIBA | T3 |
| 1 | Leaves | Control | 2.18 ± 0.83a | 2.86 ± 0.07b | 2.92 ± 0.01b | 4.78 ± 0.28b | 2.12 ± 0.05b | 2.04 ± 0.06b | 2.13 ± 0.08b | 1.34 ± 0.10a |
| | | 0.05 μM V | 3.49 ± 0.64a | 2.79 ± 0.01b | 2.96 ± 0.04b | 5.01 ± 0.09b | 2.06 ± 0.03b | 1.98 ± 0.03b | 2.13 ± 0.01b | 1.01 ± 0.08a |
| | | 0.10 μM V | 2.38 ± 0.43a | 2.82 ± 0.04b | 3.11 ± 0.06b | 5.26 ± 0.13b | 2.14 ± 0.03b | 2.12 ± 0.05b | 2.21 ± 0.05b | 1.14 ± 0.07a |
| | | 0.20 μM V | 4.19 ± 0.45a | 3.33 ± 0.03c | 4.09 ± 0.03c | 6.91 ± 0.05c | 2.75 ± 0.03c | 2.92 ± 0.06c | 2.96 ± 0.07c | 1.45 ± 0.07a |
| | | 0.40 μM V | 2.45 ± 0.15a | 1.83 ± 0.05a | 1.93 ± 0.06a | 3.32 ± 0.09a | 1.36 ± 0.03a | 1.73 ± 0.04a | 1.49 ± 0.03a | 1.26 ± 0.39a |
| | Roots | Control | 4.79 ± 1.60a | 1.51 ± 0.02e | 0.02 ± 0.001b | 0.03 ± 0.002a | 0.033 ± 0.003a | 0.039 ± 0.001bc | 0.022 ± 0.001a | 33.02 ± 0.56d |
| | | 0.05 μM V | 2.67 ± 0.95a | 1.21 ± 0.01d | 0.02 ± 0.002ab | 0.02 ± 0.003a | 0.026 ± 0.005a | 0.042 ± 0.002c | 0.025 ± 0.003a | 24.38 ± 0.85bc |
| | | 0.10 μM V | 3.32 ± 0.51a | 1.04 ± 0.01c | 0.02 ± 0.001ab | 0.04 ± 0.006a | 0.033 ± 0.002a | 0.048 ± 0.002c | 0.137 ± 0.107a | 26.14 ± 0.49c |
| | | 0.20 μM V | 4.35 ± 1.29a | 0.78 ± 0.01a | 0.02 ± 0.001a | 0.03 ± 0.001a | 0.028 ± 0.001a | 0.029 ± 0.004b | 0.026 ± 0.001a | 20.19 ± 1.23a |
| | | 0.40 μM V | 3.91 ± 0.55a | 0.87 ± 0.02b | 0.02 ± 0.001a | 0.02 ± 0.001a | 0.035 ± 0.002a | 0.014 ± 0.002a | 0.023 ± 0.001a | 21.59 ± 0.61ab |
| 2 | Leaves | Cont. + KIO₃ | 3.52 ± 0.33ab | 1.04 ± 0.01c | 0.45 ± 0.005a | 0.80 ± 0.016b | 0.38 ± 0.019a | 0.46 ± 0.010a | 0.39 ± 0.005a | 1.35 ± 0.08a |
| | | 0.05 μM V + KIO₃ | 2.49 ± 0.39a | 0.93 ± 0.01b | 0.46 ± 0.008a | 0.81 ± 0.011b | 0.37 ± 0.006a | 0.49 ± 0.037a | 0.38 ± 0.006a | 1.26 ± 0.20a |
| | | 0.10 μM V + KIO₃ | 2.25 ± 0.49a | 0.92 ± 0.01b | 0.45 ± 0.007a | 0.81 ± 0.014b | 0.43 ± 0.058a | 0.46 ± 0.004a | 0.40 ± 0.019a | 1.57 ± 0.20a |
| | | 0.20 μM V + KIO₃ | 3.87 ± 1.33ab | 1.15 ± 0.01d | 0.46 ± 0.004a | 0.72 ± 0.027a | 0.39 ± 0.007a | 0.51 ± 0.012a | 0.39 ± 0.007a | 1.44 ± 0.05a |
| | | 0.40 μM V + KIO₃ | 5.77 ± 0.46b | 0.52 ± 0.01a | 0.45 ± 0.005a | 0.81 ± 0.005b | 0.40 ± 0.012a | 0.49 ± 0.008a | 0.41 ± 0.005a | 1.82 ± 0.30a |
| | Roots | Cont. + KIO₃ | 2.45 ± 0.64a | 0.73 ± 0.01a | 0.025 ± 0.001c | 0.13 ± 0.005b | 0.028 ± 0.002a | 0.023 ± 0.002a | 0.021 ± 0.001a | 24.43 ± 0.94a |
| | | 0.05 μM V + KIO₃ | 2.18 ± 0.22a | 0.83 ± 0.02b | 0.013 ± 0.001a | 0.03 ± 0.002a | 0.034 ± 0.002a | 0.018 ± 0.003a | 0.022 ± 0.001a | 22.35 ± 1.15a |
| | | 0.10 μM V + KIO₃ | 1.94 ± 0.72a | 0.76 ± 0.02ab | 0.016 ± 0.001ab | 0.03 ± 0.002a | 0.031 ± 0.003a | 0.020 ± 0.004a | 0.021 ± 0.002a | 26.33 ± 1.38a |
| | | 0.20 μM V + KIO₃ | 1.37 ± 0.33a | 1.37 ± 0.02c | 0.019 ± 0.001b | 0.03 ± 0.002a | 0.027 ± 0.004a | 0.027 ± 0.005a | 0.022 ± 0.003a | 26.30 ± 1.73a |
| | | 0.40 μM V + KIO₃ | 2.33 ± 0.38a | 0.73 ± 0.01a | 0.031 ± 0.001d | 0.41 ± 0.004c | 0.075 ± 0.011a | 0.028 ± 0.002a | 0.039 ± 0.000a | 23.75 ± 0.89a |
| 3 | Leaves | Cont. + 5-ISA | 0.81 ± 0.17a | 1.14 ± 0.33a | 2.38 ± 0.07c | 0.20 ± 0.009a | 0.09 ± 0.002a | 0.12 ± 0.005a | 0.10 ± 0.003a | 7.50 ± 0.49a |
| | | 0.05 μM V + 5-ISA | 0.94 ± 0.26a | 1.16 ± 0.01a | 1.95 ± 0.06b | 0.26 ± 0.012b | 0.09 ± 0.005a | 0.12 ± 0.006a | 0.10 ± 0.005a | 8.30 ± 0.53a |
| | | 0.10 μM V + 5-ISA | 0.72 ± 0.25a | 1.24 ± 0.04a | 1.90 ± 0.04b | 0.23 ± 0.021ab | 0.09 ± 0.003a | 0.10 ± 0.001a | 0.10 ± 0.001a | 8.38 ± 0.15a |
| | | 0.20 μM V + 5-ISA | 1.20 ± 0.36a | 1.18 ± 0.05a | 1.24 ± 0.02a | 0.18 ± 0.009a | 0.09 ± 0.009a | 0.10 ± 0.010a | 0.10 ± 0.002a | 8.07 ± 0.46a |
| | | 0.40 μM V + 5-ISA | 0.99 ± 0.26a | 1.40 ± 0.05a | 1.92 ± 0.06b | 0.23 ± 0012ab | 0.09 ± 0.003a | 0.10 ± 0.003a | 0.10 ± 0.001a | 7.13 ± 0.36a |
| | Roots | Cont. + 5-ISA | 0.86 ± 0.13a | 16.47 ± 0.61c | 14.24 ± 0.24a | 30.3 ± 0.26a | 0.07 ± 0.005b | 0.03 ± 0.004ab | 0.03 ± 0.004a | 9.70 ± 0.18b |
| | | 0.05 μM V + 5-ISA | 2.54 ± 0.55b | 8.80 ± 0.13a | 28.18 ± 0.33e | 31.0 ± 0.60a | 0.06 ± 0.001a | 0.03 ± 0.003ab | 0.02 ± 0.002a | 6.47 ± 0.07a |
| | | 0.10 μM V + 5-ISA | 2.30 ± 0.36ab | 10.37 ± 0.40b | 16.34 ± 0.73b | 42.8 ± 2.08c | 0.07 ± 0.002ab | 0.05 ± 0.003b | 0.02 ± 0.001a | 6.48 ± 0.73a |
| | | 0.20 μM V + 5-ISA | 1.84 ± 0.09ab | 11.12 ± 0.15c | 21.34 ± 0.27d | 36.1 ± 0.18b | 0.07 ± 0.001ab | 0.03 ± 0.002a | 0.03 ± 0.007a | 7.72 ± 1.08ab |
| | | 0.40 μM V + 5-ISA | 1.52 ± 0.45ab | 8.11 ± 0.14a | 18.79 ± 0.30c | 35.6 ± 0.21b | 0.07 ± 0.003ab | 0.03 ± 0.005a | 0.03 ± 0.002a | 6.10 ± 0.46a |

**Table 7.** *Cont.*

| Exp. No. | Part of Plants | Treatments | (mg·kg⁻¹ D.W.) | | | | | | | |
|---|---|---|---|---|---|---|---|---|---|---|
| | | | BeA | SA | 5-ISA | 3,5-diISA | 2-IBeA | 4-IBeA | 2,3,5-TIBA | T3 |
| 4 | Leaves | Cont. + 3,5-diISA | 1.22 ± 0.37a | 0.64 ± 0.03c | 0.24 ± 0.004a | 10.46 ± 0.41c | 0.08 ± 0.007a | 0.09 ± 0.004a | 0.09 ± 0.003a | 10.52 ± 0.24a |
| | | 0.05 µM V + 3,5-diISA | 0.83 ± 0.19a | 0.69 ± 0.01c | 0.26 ± 0.002ab | 6.28 ± 0.06a | 0.09 ± 0.007a | 0.08 ± 0.005a | 0.10 ± 0.005ab | 11.12 ± 0.27ab |
| | | 0.10 µM V + 3,5-diISA | 0.98 ± 0.15a | 0.53 ± 0.01b | 0.31 ± 0.005b | 7.62 ± 0.26b | 0.09 ± 0.004a | 0.09 ± 0.008a | 0.11 ± 0.005ab | 13.36 ± 0.73c |
| | | 0.20 µM V + 3,5-diISA | 1.80 ± 0.63a | 0.41 ± 0.02a | 0.27 ± 0.004c | 8.07 ± 0.04b | 0.10 ± 0.003a | 0.09 ± 0.002a | 0.11 ± 0.005ab | 14.44 ± 0.37c |
| | | 0.40 µM V + 3,5-diISA | 1.64 ± 0.36a | 0.55 ± 0.01b | 0.35 ± 0.011d | 11.38 ± 0.28c | 0.10 ± 0.010a | 0.11 ± 0.011a | 0.12 ± 0.007b | 12.96 ± 0.52bc |
| | Roots | Cont. + 3,5-diISA | 1.33 ± 0.43a | 5.91 ± 0.08c | 3.34 ± 0.04a | 701.8 ± 2.31a | 0.03 ± 0.001b | 0.013 ± 0.002b | 0.003 ± 0.0006ab | 8.20 ± 0.93a |
| | | 0.05 µM V + 3,5-diISA | 2.54 ± 0.41a | 6.68 ± 0.03d | 4.99 ± 0.02c | 712.4 ± 4.58a | 0.02 ± 0.001a | 0.011 ± 0.001ab | 0.005 ± 0.0001b | 9.41 ± 0.89a |
| | | 0.10 µM V + 3,5-diISA | 1.54 ± 0.41a | 6.79 ± 0.06d | 3.07 ± 0.04a | 694.4 ± 11.33a | 0.01 ± 0.001a | 0.009 ± 0.001ab | 0.003 ± 0.0004ab | 10.61 ± 0.25a |
| | | 0.20 µM V + 3,5-diISA | 1.88 ± 0.40a | 5.54 ± 0.02b | 3.20 ± 0.02a | 716.7 ± 5.02a | 0.01 ± 0.001a | 0.007 ± 0.0008ab | 0.001 ± 0.0003a | 9.69 ± 0.79a |
| | | 0.40 µM V + 3,5-diISA | 1.80 ± 0.77a | 5.24 ± 0.06a | 3.72 ± 0.13b | 681.9 ± 14.15a | 0.04 ± 0.001c | 0.005 ± 0.0006a | 0.002 ± 0.0005ab | 8.89 ± 0.38a |

Means in the column followed by different letters separately for each experiment differ significantly at $p < 0.05$ ($n = 8$).

## 4. Discussion

### 4.1. Biomass Production and Vanadium Accumulation in Lettuce Plants

In the present study, the application of vanadium into the nutrient solution had no effect on the biomass of lettuce roots and leaves. Even the application of the highest dose of vanadium (0.4 µM V) did not cause any visible symptoms of its toxicity to plants. This clearly suggests that the applied doses were on the safe level for lettuce plants. Welch and Huffman [17] did not note any significant effect of 1 µM V (as $NH_4VO_3$) on the yield of lettuce and tomato as compared to the control combination (characterized by a trace amount of vanadium, i.e., <0.07 µM V).

The studies by Vachirapatama et al. [40] revealed the possibility of increasing root-to-leaf transfer of vanadium when very high, potentially lethal doses of that element are applied to plants. In the plants of Chinese green mustard, the high content of vanadium in the nutrient solution between 0.39 and 1.57 mM V caused a proportional increase in vanadium content in respective plant organs according to the order: roots > steams > leaves. Importantly, roots contained a few hundred times more V than leaves and stems [40]. Similar relations were found in the plants of sweet basil fertilized with $NH_4VO_3$ in doses of 0.1, 0.1, 0.39, and 0.79 mM V [41]. It was also revealed that fertilization with 1 µM V (as $NH_4VO_3$) proportionally increased the content of vanadium in the leaves and roots of lettuce and tomato [17].

In the present studies, increasing vanadium doses caused a gradual increase in its content only in the roots (Figure 2). The obtained results are in agreement with observations made on various crops, including tomato and Chinese green mustard [40], soybeans [19], rice [42] and lettuce [43]. In these studies, vanadium accumulated in roots rather than in above-ground parts of the plant. It needs to be mentioned that the applied doses between 0.05 and 0.40 µM V, apart from being safe for plants, may have been too low to observe an efficient root-to-leaf distribution of that element. Vanadium doses exceeding 0.79 mM V impaired the growth and development of tomato and Chinese green mustard plants [40]. A toxic dose of vanadium (as $VOSO_4$) for soybean plants was 1.2 mM V[19]. In the case of rice plants, vanadium toxicity was revealed for the 0.39 mM V dose [42]. When excessive vanadium concentrations were applied, plastid degradation occurred in maize and horse bean plants [43]. Gil et al. [44] observed a 15% decrease in lettuce biomass, even for the lowest dose of 0.002 mM V (as $NH_4VO_3$) and the application of 0.02 mM V reduced plant biomass by approximately 64%. Moreover, root darkening, a decrease in the number of secondary roots, and a loss of leaf turgidity was observed.

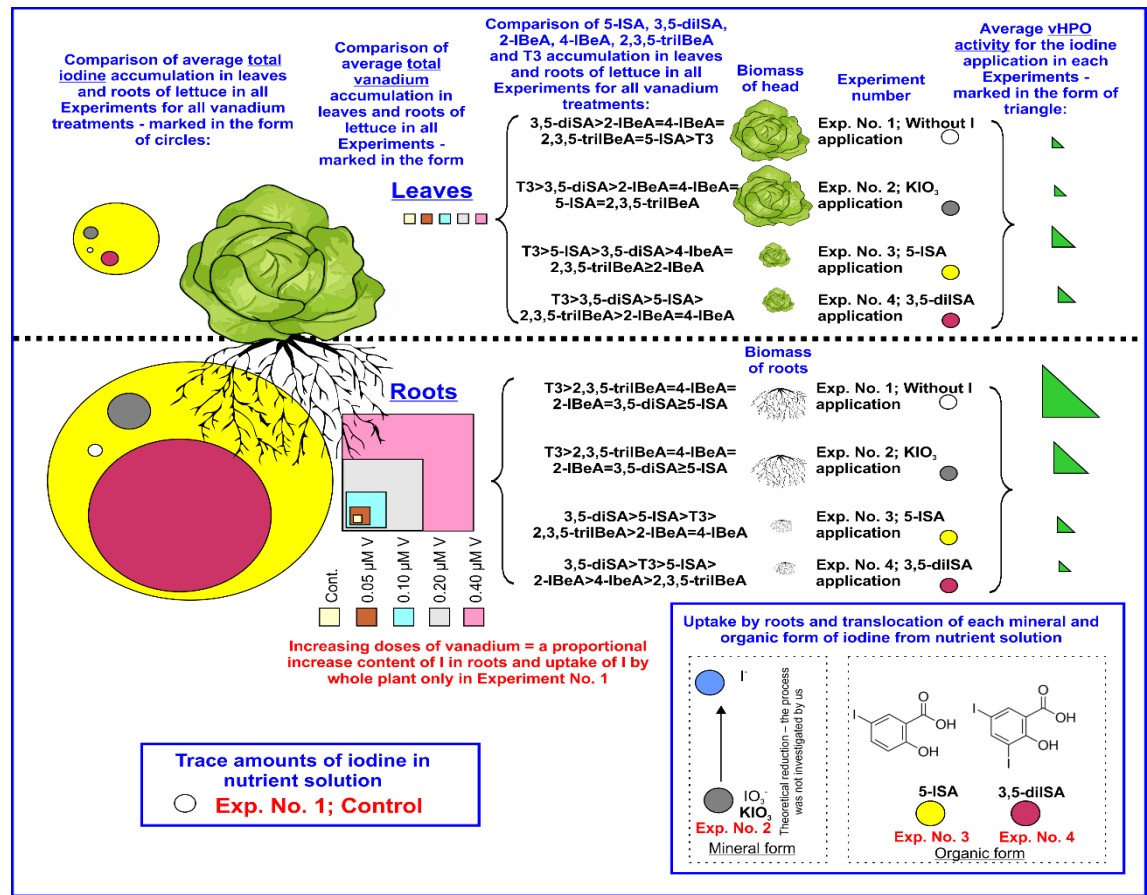

**Figure 2.** Summary of the study.

### 4.2. Iodine Accumulation vs. Vanadium Application and vHPO Activity

The enzyme vanadium-dependent haloperoxidase (vHPO) that is present in marine alga *Laminaria digitata* participates in the process of iodine uptake/release from cells with hypoiodous acid (HIO) as an intermediate [27]. An indirect role of vHPO in the volatilization of elemental iodine $I_2$ by marine algae is related, among others, to plant response to oxidative stress. Furthermore, the release of volatile methyl iodide also occurs in marine algae [4] and higher plants such as *Arabidopsis thaliana*, rice [45,46], or lettuce [47].

In the analyzed literature, no information can be found on the interaction between vanadium application and the level of iodine accumulation in crop plants. The effect of vanadium on the activity of vHPO in crop plants also has not been studied. Basically, the description of the structure, activity, and role of vHPO rely on the results obtained for enzymes isolated from various species of marine algae. Considering the structure and functioning of vHPO, it is classified into the group of histidine phosphatase/peroxidase super family [48]. Studies conducted by Colin et al. [49] revealed that in vitro activity of vHPO isolated from *Laminaria digitata* increased when KI was applied in the range of 0–10 mM I and dropped for KI applied in doses >20 mM I.

The trial was undertaken to determine the enzymatic activity of vHPO in lettuce with no previous analytical protocol and based on the assumption that lettuce extracts may exhibit enzymatic activity typical for vHPO. The assayed activity of vHPO is a total activity of vanadium-dependent peroxidases in plant extract and is a derivative of interaction between iodine and vanadium or other halogens. The results of the study indirectly indicate the functioning of various mechanisms regulating the activity of vHPO in lettuce, depending on the application of vanadium and different iodine compounds (KIO_3, 5-ISA and 3,5-diSA).

A significant increase in vHPO activity after the application of vanadium was noted only in the leaves of plants non-fertilized with iodine(Experiment No. 1) and was positively correlated with iodine content in roots ($r^2 = 0.78$*) and leaves ($r^2 = 0.79$*).These results suggest that vanadium fertilization may improve via increasing vHPO activity, the process of iodine uptake, and accumulation in lettuce roots and leaves only when plants are cultivated in the presence of a trace amount of that element in the nutrient solution (Experiment No. 1). Application of 3,5-diISA reduced vHPO activity in lettuce roots to levels lower than those noted for 5-ISA. At the same time, application of the highest dose of V together with 3,5-diISA decreased the activity of vHPO in leaves the most. That observation was accompanied by a slight increase in T3 content in leaves, but no effect on leaf accumulation of SA, BeA, 3,5-diSA, 5-ISA, 2-IBeA, 4-IBeA and 2,3,5-triIBeA was observed.

The reduction in the vHPO activity in roots of plants treated with 5-ISA and 3,5-diISA suggests that these iodosalicylates are taken up by the roots by different mechanisms than those related to iodide uptake with the participation of vHPO as described in marine alga. It is worth mentioning that iodates ($IO_3^-$) undergo reduction to $I^-$ in the root zone most probably by a specific reductase [50] or, alternatively, nitrate reductase [4].

### 4.3. SA and Iodosalicylate Metabolism

In the plant organisms, benzoic acid (BeA) is a precursor of SA, the latter being considered as a signaling molecule and plant growth regulator [51]. On the other hand, 2,3,5-triiodobenzoic acid (2,3,5-triIBeA) plays a role of auxin inhibitor [52]. Van de Wouwer et al. [53] revealed that *p*-iodobenzoic acid (synonym: 4-IBeA) and its derivatives inhibit the process of lignification by reducing the activity of cinnamate 4-hydroxylase—a key enzyme in the phenylopropanoid pathway leading to the synthesis of lignin polymers. Crisan [54] revealed that exogenous 3-iodobenzoic acid (3-IBeA—its content was not determined in our study) stimulated root elongation and the formation of adventitious roots.

Previous studies on tomato plants showed that plant preference towards the uptake and accumulation of 5-ISA and 3,5-diISA in leaves and roots depended on the growth stage of plant: stage of 5–6 true leaves [26], intensive vegetative growth, and fruiting [39]. In the present studies, iodine applied as 5-ISA was taken up and distributed more easily than 3,5-diISA.

It was revealed that 3,5-diSA, 5-ISA, 2-IBeA, 4-IBeA, and T3 are present and synthesized in lettuce plants and, to our knowledge, this is a first report of that matter. In the case of **PDTHA** (Plant Derived Thyroid Hormone Analogs), its synthesis in plant tissues has been previously hypothesized by Lima et al. [55]. The exact pathway of PDTHA synthesis is yet to be described. It is worth mentioning that the first report of PDTHA was presented by Fowden [56], who revealed that, after fertilization with iodine, the plants of astra and *Salicornia* sp. contained (and therefore synthesized): 3,5-diiodothyrosyne, 3,5-diiodothyronine and 3,5,3'-triiodothyronine. Bean and barley plants grown in the presence of iodine contained only 3,5-diiodothyrosyne. A specific enzymatic system is required for the production of PDTHA in plants. Fenical [57] informed that the enzymatic process of halogenation, i.e., the incorporation of iodine (or other halogens), was described for mushrooms. Furthermore, the presence of 3-iodothyrosyne, 2,5-diiodothyrosyne, and 3,5,3'-triiodothyronine has been confirmed in *Rhodophyta* algae [57]. To our knowledge, the process of synthesis and metabolism of iodosalicylates and T3 in higher plants has not yet been described. Based on quantitative relations between analyzed iodosalicylates and iodobenzoates in the roots and leaves of lettuce from all four experiments, a hypothetical overview of that process can be proposed (Figures 2 and 3).

It needs to be underlined that the synthesis and metabolism of the analyzed compounds in lettuce plants varied depending on the form of applied iodine rather than the dose of vanadium. In all four experiments, the highest content of 2,3,5-triIBeA, 5-ISA, 2-IBeA and 4-IBeA was measured in the leaves of control plants, i.e., not fertilized with iodine (Experiment No. 1). On the other hand, the content of 3,5-diSA in the leaves of the control plants was slightly lower than after the application of 3,5-diSA (Experiment No. 4). It seems that the basic iodine metabolites in the plants grown in the presence of trace amounts of iodine were iodosalicylates, iodobenzoates, and T3. However, the obtained results do

not provide the background for the possible physiological function of these organoiodine compounds in lettuce plants.

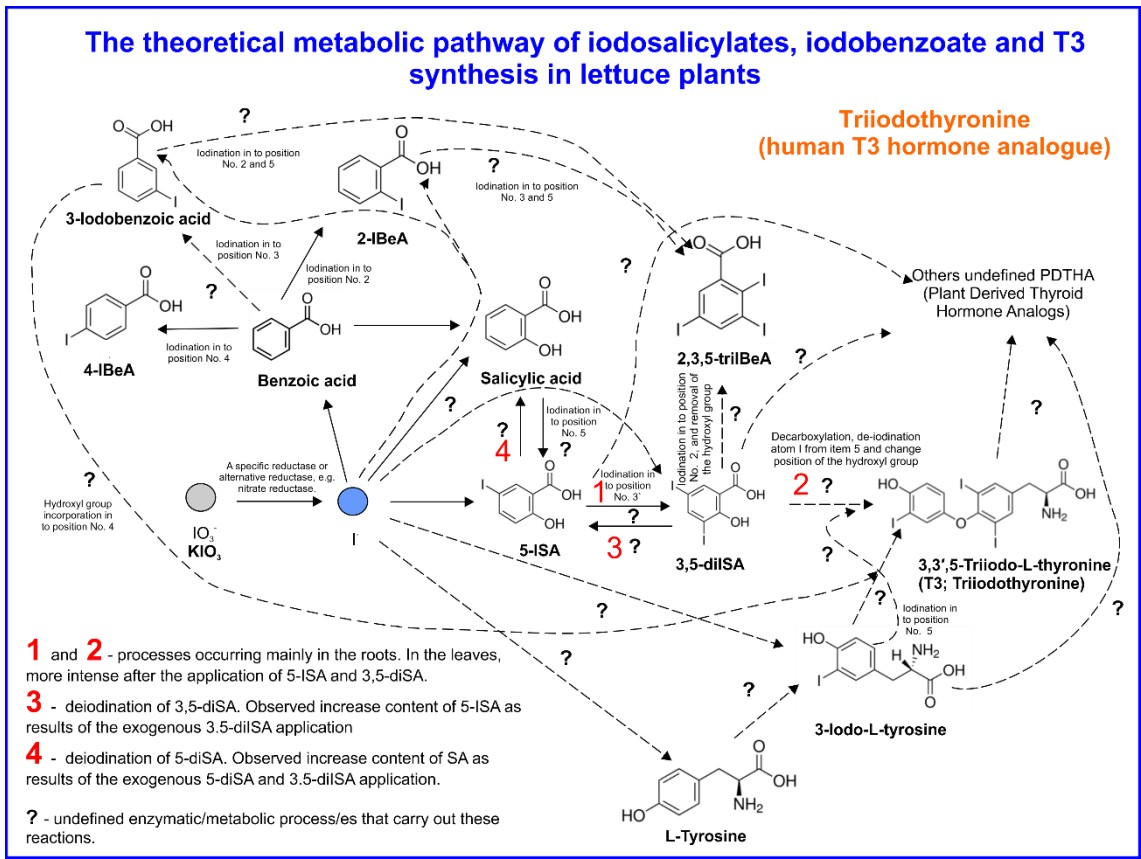

**Figure 3.** Theoretical metabolic pathway of iodosalicylates and iodobenzoates and T3 synthesis in lettuce plants.

The application of $KIO_3$ into the nutrient solution decreased the level of iodosalicylates and iodobenzoates in lettuce leaves. Most likely, in the case of increased accumulation of inorganic iodine, different pathways of its metabolism were activated (i.e., including iodine volatilization through methylation) than those engaged in the synthesis of iodosalicylates and iodobenzoates. Based on these results, it can be concluded that, after the application of iodosalicylates, endogenous iodobenzoates (2-IBeA, 4-IBeA and 2,3,5-triIBeA) as well as 3,5-diISA and 5-ISA are degraded, conversed into other compounds or volatilized in methyl forms. One of the formed compounds includes T3 as its level was higher in lettuce leaves fertilized with 5-ISA and 3,5-diISA than in the control and $KIO_3$ plants.

Taking the above into consideration, this supports the proposed description of possible synthesis of T3 and the metabolism of exogenous iodosalicylates in lettuce plants (Figure 3). Biosynthesis of T3 is independent of applied vanadium dose as well as of vHPO activity and most likely occurs in roots. The transport of T3 from roots to leaves is strongly limited or alternatively; T3 is conversed in leaves into other compounds from the PDTHA group.

Plant fertilization with $KIO_3$ did not modify the level of T3 in leaves and roots of lettuce, which suggests that the content of T3 in lettuce leaves remains stable for a trace and increased concentration of inorganic iodine in the root zone and plant tissues. In the conditions of increased concentrations of 5-ISA and 3,5-diSA in plants, an efficient distribution of both iodosalicylates into the leaves was observed. This was followed by a substantial increase in the T3 level in leaves and a decrease of T3 content in roots as compared to plants from the control and $KIO_3$ experiments. Therefore, it can be concluded that 5-ISA was converted into 3,5-diISA in both leaves and roots and the latter compound

may have been utilized for the synthesis of T3 through, for instance, its joining with 3-iodo-L-tyrosine or other metabolic pathways that are presented in Figure 3.

A slight decrease in the biosynthesis of BeA was also observed in the roots of plants grown in the presence of $KIO_3$ as compared to the control plants (Experiment No. 2 versus Experiment No. 1). Plant fertilization with $KIO_3$ also contributed to a simultaneous decrease of the content of (a) SA in roots and leaves; and(b) 5-ISA, 3,5-diSA, 2-IBeA, 4-IBeA, and 2,3,5-triIBeA in leaves. These results indicate that different pathways of iodine metabolism were activated by $KIO_3$applicationthat was directed, among others, on iodine methylation. In the case of plant fertilization with 5-ISA and 3,5-diISA, iodine metabolism was directed to the synthesis of T3 or other organic iodine compounds including those classified as PDTHA. This may have been a direct cause of obtaining lower biomass after plant treatment with iodosalicylates. In addition, the content of 5-ISA, 3,5-diSA, and T3 in the leaves of plants treated with iodosalicylates was higher than the physiological level noted in the control plants. This could have been another factor that modified the functioning of phytohormones related to the growth and development of lettuce plants fertilized with iodosalicylates.

Furthermore, the obtained results indirectly indicate that the exogenous 5-ISA and 3,5-diSA weakened the synthesis of 2-IBeA, 4-IBeA, and 2,3,5-triIBeA in lettuce plants. A significantly higher content of SA as well as a decreased content of BeA in the roots of plants fertilized with 5-ISA and 3,5-diISA suggest that some share of iodosalicylates, apart from T3/PTDHA synthesis, could have undergone the process of de-iodination into SA (Figure 3).

## 5. Conclusions

Within the range of applied doses, vanadium had no influence on lettuce growth. Application of 5-ISA and 3,5-diISA resulted in a higher total level of iodine accumulation and uptake by plants (leaves+roots) as compared to $KIO_3$. At the same time, the content of iodine reached the toxicity value as the plants were characterized by approximately nine- and seven-times lower biomass as compared to the control and $KIO_3$ plants.

The enzymatic system of vHPO was engaged in the uptake of inorganic iodine forms by lettuce plants, particularly for trace amounts of $I^-$ in the root environment as well as those formed after the reduction of $IO_3^-$. Iodosalicylates: 5-ISA and 3,5-diISA underwent various mechanisms regulating the uptake, distribution and metabolism processes in plants that were independent of vHPO functioning.

The values of 5-ISA and 3,5-diISA analysis in the control plants indicate that these compounds are physiologically present in lettuce. It seems that exogenous application of iodosalicylates may increase the plant ability to synthesize organoiodine compounds containing iodine bound into the aromatic group.

After plant fertilization with 5-ISA and 3,5-diISA, iodine metabolism in plants was directed into the synthesis of T3 and, most likely, other organoiodine compounds, including PDTHA. The application of exogenous $KIO_3$ activated other metabolic pathways in lettuce plants, probably including iodine methylation.

In the presence of trace amounts of iodine as well as for iodine applied as $KIO_3$, the processes of iodosalicylates conversion into T3 were more efficient in roots than in leaves. In the case of exogenous application of iodosalicylates, the increase of T3 biosynthesis in leaves was related to improved transport of 3,5-diISA rather than 5-ISA. Increased uptake of iodosalicylates also reduced the biosynthesis of BeA, which is a precursor of SA. On the other hand, some share of taken up 5-ISA and 3,5-diISA could have been degraded or conversed into SA through elimination of one or two molecules of iodine. The hypothetical activation of that metabolic pathway only in the roots of plants fertilized with iodosalicylates could have been caused by a significant increase of T3 content in leaves. An increase in SA level in lettuce from the combinations with 5-ISA and 3,5-diISA could have been an effect of increased synthesis of SA as a response to the stress effect exerted by 5-ISA and 3,5-diISA.

**Author Contributions:** Methodology S.S.; Formal analysis, S.S. and I.L.-S.; Funding acquisition, S.S.; Investigation, S.S.; I.K.; M.H.; I.L.-S.; M.G., Ł.S.; M.C. and J.P. Resources, S.S. and I.K.; Supervision, S.S. and I.K.; Visualization, S.S.

and I.L.-S.; Writing—original draft, S.S.; Writing—review and editing, S.S.; I.K., I.L.-S.; Ł.S. and M.C. All authors have read and agreed to the published version of the manuscript.

**Funding:** This research was financed by the National Science Centre, Poland (Grant No. UMO-2017/25/B/NZ9/00312).

**Acknowledgments:** The method of enriching plants with iodine with: 5-iodosalicylic acid and 3,5-diiodosalicylic acid is patented by the Polish Patent Office—patent number P.410806 (20 November 2017).

**Conflicts of Interest:** The authors declare no conflict of interest.

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
