# Peer review of "SelectedAspects of Iodate and Iodosalicylate Metabolism in Lettuce Including the Activity of Vanadium Dependent Haloperoxidases as Affected by Exogenous Vanadium"

_agronomy, doi:10.3390/agronomy10010001_

Round 1

Reviewer 1 Report

Delete this: "The results of pH measurement were used to regulate the pH nutrient solution with the use of 38% nitric acid" (because is a repetition).

Reviewer 2 Report

Authors have done a great effort in order to justify the questions and to improve the text according to the comments and suggestions. Their explanations are satisfied for me, and I have not more suggestions to the manuscript.

This manuscript is a resubmission of an earlier submission. The following is a list of the peer review reports and author responses from that submission.

Round 1

Reviewer 1 Report

Thank you for giving me the opportunity to correct and evaluate this article. The paper is very interesting but the authors do not write how long the experiments lasted, and it is not clear if the four experiments took place simultaneously (correctly) or at different times. The results described are many and often difficult to understand. For the reasons listed above, I advise the authors to reduce the number of results presented and to highlight the most important results. Annotations: 1) the Table 1 is difficult to read; 2) the Figure 1 has no bar to evaluate the dimensions 3) Iodine uptake in Table 3 is not described 4) not always the  reference number is correct (for example: 40,44, 49, etc)

Reviewer 2 Report

In the manuscript entitled "Selected aspects of iodate and iodosalicylate metabolism in lettuce including the activity of vanadium dependent haloperoxidases as affected by exogenous vanadium"

Line 36: NFT please write a full name?

Line 39: On which basis authors choose 10 µM KIO3?

Line 37, 39 and 40 use brackets instead of /…../?

Rewrite the whole abstract in a more significant manner.

Keywords should be arranged alphabetically

Write a brief introduction in a more specific way?

Line 199 writes ‘we’ instead of ‘wee’

Results

Table 2, why experiment 3 is non-significant each other’s in leaf biomass, root biomass, and whole plant biomass?

What is the significance of these results?

Some results are very confusing. Please revise the whole results carefully in a more significant way.

Overall, the topic is interesting and certain results presented are useful.

Reviewer 3 Report

I suggest reducing the discussion and describing some parts of the manuscript more clearly:
1) In the abstract it is necessary to insert the effect that the experimental treatments had on the production of the lettuce, of what type of lettuce it is and that the experiment was conducted in a greenhouse.
2) In materials and methods it is necessary to specify the electrical conductivity value of the nutritive solutions and how the pH was controlled.
3) In table 1, what does "Heating temperature" mean? 10 ° C by day and 16 ° C by night?
4) In materials and methods it should be indicated when the experiments were carried out and how long the crop cycles lasted.
5) You have not indicated the experimental design adopted. How were the 4 replications arranged? I fear that the arrangement of treatments and replications has not been done correctly (in an experimental design).
6) Figures 2A, 2B and 2C are not legible (it is not necessary to have such a large ordinate axis). I consider it opportune to transform figure 2 and figure 3 in tables.

It is necessary to correct ml in mL and the decimal values ​​(with the dot and not with the comma).
The writing of the results can be improved. For example, "In the present study, application of increasing doses of vanadium: 0.05, 0.1 and 0.2 μM. V into the nutrient solution had no effect on the biomass of bed roots and leaves as compared to control without V application (Fig. 4)." it can be simplified as follows: "In the present study of application of more vanadium into the nutrient solution had no effect on the biomass of lettuce roots and leaves (Fig. 4)."
